# Controllable Representation Learning for Time-series Analysis

## ABSTRACT

Representation learning for time series typically relies on reliable anchors: smooth input signals or dense supervision that constrain latent dynamics. When both are degraded due to noise, missing values, or irregular sampling, hidden states will drift and standard methods will collapse. To tackle this problem, we propose a conceptual shift: treating representation learning itself as a control problem. Our framework, Neural Feedback Control (NFC), actively regulates latent trajectories using confidence-weighted pseudo-observations and pseudo-labels, combining pseudo data-based controllers with continuous-time dynamics and residual-based feedback. This design transforms latent space evolution from passive inference into a controllable process. In contrast to Neural ODEs/CDEs, which model latent dynamics without stability guarantees, and predictive coding approaches that propagate errors without explicit contraction control, NFC provides a feedback-driven mechanism with provable stability under partial observability. Theoretically, we prove that under mild conditions, NFC guarantees exponential decay of output error to a bounded region, providing a certified stability guarantee. Every module in NFC (pseudo-signal generation, confidence weighting, and feedback penalties) plays a role in a single closed-loop control system, transforming representation learning into active regulation rather than passive inference. Empirically, NFC achieves substantial robustness gains: over 50% lower forecasting error on power load datasets and more than 10% higher accuracy on human activity dataset with 30% missing data. These results highlight task-aware latent control as an effective approach for stabilizing representation learning when conventional anchors fail.

## 1 INTRODUCTION

Learning meaningful representations from time series underpins advances in domains such as energy Li et al. (2025), transportation Nguyen et al. (2018), and healthcare Rubanova et al. (2019). Recent approaches often integrate dynamical modeling Chen et al. (2018); Kidger et al. (2020) to capture temporal dependencies, and they rely on strong supervision or high-quality inputs to constrain latent states. These anchors, such as smooth input signals or dense and reliable labels, stabilize the hidden dynamics during training and prevent collapse. However, in many real-world settings, both anchors fail simultaneously: sensor readings are noisy, irregular, or missing Che et al. (2018); Rubanova et al. (2019), while supervision is sparse or weakly aligned with the underlying dynamics. Without a trustworthy anchor, hidden states drift, features become unstable, and downstream performance deteriorates. This challenge motivates a rethinking of how representation learning should proceed when neither inputs nor labels can serve as reliable guides.

We propose a conceptual shift: representation learning should not be treated as passive inference, but as an *active control problem* in the latent space. Instead of assuming that useful features will emerge from corrupted inputs and labels, we treat latent states as dynamical variables whose trajectories must be actively regulated. To realize this principle, we introduce **Neural Feedback Control (NFC)**. NFC generates confidence-weighted pseudo-observations and pseudo-labels and integrates them as control inputs to stabilize latent trajectories. This reframes the evolution of latent space from a process constrained only by data fidelity into one explicitly guided by feedback. By combining the robustness of control theory with the flexibility of deep representation learning LeCun et al. (2015); Bronstein et al. (2021), NFC provides a foundation for robust learning under degraded supervision.

A distinguishing advantage of this control-theoretic perspective is that it enables optimization over full latent trajectories rather than isolated outputs. Standard methods typically minimize error only at terminal predictions—forecasts at the end of a horizon or logits at classification time. When supervision is sparse or noisy, these terminal losses provide weak or misleading signals, and latent drift accumulates unchecked. NFC instead performs differentiable rollouts of latent states across time, exposing the entire trajectory to gradient-based optimization van den Oord et al. (2018). This trajectory-level view allows the model to propagate supervision backward across multiple steps, enabling *multi-step credit assignment* Bengio et al. (1994) that improves alignment even when intermediate labels are absent. Moreover, the unrolled trajectories serve as a *diagnostic lens*: by explicitly modeling how errors compound over time, NFC learns to correct instabilities in real time, mitigating exposure bias Bengio et al. (2015) and producing representations that remain smooth, task-relevant, and robust to corruption. While Neural ODEs (Chen et al., 2018) and Neural CDEs (Kidger et al., 2020) learn expressive continuous dynamics, they lack mechanisms to prevent drift under weak supervision. Predictive coding frameworks (Whittington & Bogacz, 2017) incorporate error feedback, but do not provide stability guarantees or integrate uncertainty-aware pseudo-signals. NFC unifies these perspectives by embedding pseudo-signal generation, confidence weighting, and closed-loop feedback into a control-theoretic framework, creating principled guarantees and practical robustness.

This trajectory-centric design also unifies the roles of input reconstruction, output alignment, and task relevance. In conventional pipelines, the encoder, transition model, and decoder are trained in modular or sequential fashion. Under degraded supervision, such separation fails: the encoder overfits to noise, the decoder chases sparse labels, and the latent dynamics drift without regulation. NFC instead couples all components (encoder, latent controller, and decoder, through feedback) ensuring that every latent adjustment is shaped by its impact on the downstream task. Concretely, pseudo-observations and pseudo-labels act as control signals that steer hidden states; their confidence weights attenuate unreliable cues while amplifying consistent ones. A residual-based feedback loop updates these pseudo-signals by monitoring prediction errors when ground truth is available, closing the loop between dynamics and supervision. This end-to-end optimization makes latent dynamics a negotiation space between noisy inputs and weak labels, moderated by task performance.

Beyond design intuition, NFC is supported by a rigorous theoretical analysis. By treating the latent dynamics as a controlled system, we establish conditions under which the output error decreases exponentially and converges to a bounded region. Specifically, under mild assumptions on model smoothness, readout well-conditioning, and bounded disturbances, we prove that the tracking error decays at an exponential rate, with the residual bound scaling linearly in disturbance and mismatch levels. The constants in this bound can be estimated from the trained model, e.g., via Jacobian sensitivity metrics, making the certificate practical to evaluate. Importantly, the guarantee extends to hybrid ODE–RNN backbones under a mild non-expansive jump condition, providing robustness even in irregularly sampled or partially observed settings. Such guarantees distinguish NFC from prior neural dynamical models Chen et al. (2018); Rubanova et al. (2019); Kidger et al. (2020), which offer expressive latent dynamics but lack principled mechanisms to prevent drift. We show in experiments that the theoretical guarantees enable faster convergence than all other methods.

Empirically, NFC demonstrates substantial robustness gains across both forecasting and classification tasks. On energy forecasting benchmarks, NFC reduces error by more than 50% compared to ODE–RNN, highlighting its ability to stabilize dynamics when inputs are corrupted. On classification tasks with 30% missing data, NFC improves accuracy by over 10 points on human activity recognition and similar margins on vehicle classification, showing that trajectory-level feedback can recover discriminative features even under severe sparsity. Beyond accuracy, NFC also provides better uncertainty calibration Guo et al. (2017); Lakshminarayanan et al. (2017), reflecting its feedback-driven capacity to assess and correct latent instabilities. These results indicate that task-aware latent control is both theoretically sound and practically impactful across diverse domains.

In summary, this work contributes a new paradigm for representation learning under degraded supervision. (1) We frame latent representation learning as an active control problem, introducing the principle of regulating hidden trajectories with feedback. (2) We develop Neural Feedback Control (NFC), which integrates confidence-weighted pseudo-observations and pseudo-labels as control inputs to stabilize latent states. (3) We provide a theoretical guarantee that NFC ensures exponential decay of output error toward a bounded region under mild assumptions, yielding a certified stability certificate. (4) We validate NFC extensively, showing large and consistent improvements across

forecasting and classification under noise, sparsity, and irregular sampling. Importantly, these elements are not isolated tricks: pseudo-signals act as reference generators, confidence weights as adaptive gains, and feedback penalties as stability corrections, all unified in a closed-loop control formulation (Fig. 1, Alg. 1). Together, these contributions establish *task-aware latent control* as a general mechanism for stabilizing representation learning when conventional anchors fail.

## 2 PROBLEM FORMULATION AND PRELIMINARIES

**Problem setting.** We consider time series with irregular sampling, noise, and partially missing supervision. Formally, we observe $\{\boldsymbol{x}_i\}_{i=1}^N$ at timestamps $\{t_i\}_{i=1}^N$, together with labels $y$ that may be incomplete. We introduce binary masks $m_i^x, m_i^y \in \{0, 1\}$ to denote availability (1 = observed), and generate *composite signals* $\boldsymbol{x}_i^\star = m_i^x \boldsymbol{x}_i + (1 - m_i^x)\tilde{\boldsymbol{x}}_i$, $y_i^\star = m_i^y y + (1 - m_i^y)\tilde{y}_i$, where $(\tilde{\boldsymbol{x}}_i, \tilde{y}_i)$ are pseudo-observations and pseudo-labels produced by a generative module (See Sec. 3.1). Our goal is to train a classifier or regressor that produces accurate predictions and calibrated uncertainty, even under severe missingness. For clarity, we will use the following notations throughout: hidden state $\boldsymbol{h}(t)$, control input $\boldsymbol{u}(t)$, feedback signals $\boldsymbol{e}(t)$, and composite signals $(\boldsymbol{x}^\star, y^\star)$. A complete symbol table is given in Appendix 5. Figure 1 shows NFC framework.

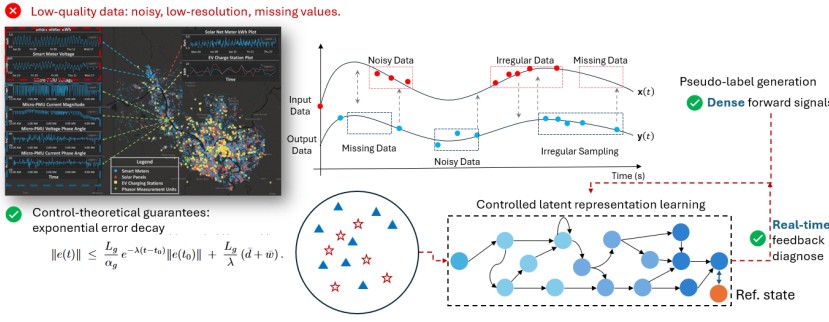

Figure 1: Overview. Pseudo-signals & confidence weights regulate latent trajectories via feedback.

**Continuous-time dynamics.** Following Neural ODEs (Chen et al., 2018; Kidger et al., 2020), we model latent dynamics by $\frac{d\boldsymbol{h}(t)}{dt} = f_\phi(\boldsymbol{h}(t), t)$, where $\boldsymbol{h}(t)$ is the hidden state and $f_\phi$ a neural network. To incorporate external modulation, we extend to $\frac{d\boldsymbol{h}(t)}{dt} = f_\phi(\boldsymbol{h}(t), \boldsymbol{u}(t), t)$, where $\boldsymbol{u}(t)$ denotes the control input. Existing variants specify $\boldsymbol{u}(t)$ using derivatives of interpolated paths (Kidger et al., 2020) or augmented states (Dupont et al., 2019). In our formulation, $\boldsymbol{u}(t)$ aggregates temporal memory, composite pseudo-signals, and their confidence weights, enabling the ODE flow to respect both dynamics and uncertainty, as described in Section 3.

## 3 NEURAL FEEDBACK CONTROL FRAMEWORK

We propose a highly expressive and theoretically grounded model, Neural Feedback Control (NFC). The overall framework is shown in Fig. 1. NFC integrates forward predictive control with feedback correction. These signals drive the evolution of a continuous hidden feature state $\boldsymbol{h}(t)$.

### 3.1 CLOSED-LOOP LATENT DYNAMICS

**Forward control via VAE-generated pseudo-signals.** As shown in the top right of Fig. 1, we introduce a novel VAE to create pseudo-labels for dense control signals. At step $i$, a variational encoder summarizes the masked history into a context $\mathcal{C}_i$ and infers a sequence-level latent $\xi_i$ with posterior $q_\phi(\xi_i \mid \mathcal{C}_i) = \mathcal{N}(\mu_i^\xi, \Sigma_i^\xi)$. A time-aware decoder $G_\theta$ maps $\xi_i$ to Gaussian predictions over a look-ahead horizon $M$ for pseudo-observations and a pseudo-label: $(\tilde{\boldsymbol{x}}_{i:i+M}, \Sigma_{i:i+M}^x)$, $(\tilde{y}_i, \Sigma_i^y) = G_\theta(\xi_i)$, where $\tilde{\boldsymbol{x}}_{i:i+M} = (\tilde{\boldsymbol{x}}_i, \ldots, \tilde{\boldsymbol{x}}_{i+M})$, $\Sigma_{i+k}^x$ is the (co)variance for $\tilde{\boldsymbol{x}}_{i+k}$, and $\Sigma_i^y$ is the variance for $\tilde{y}_i$. Posterior uncertainties are converted to confidences via a monotone inverse-uncertainty mapping; for instance, $c_{i+k}^x = \exp(-\tau_x \operatorname{tr} \Sigma_{i+k}^x)$, $c_i^y = \exp(-\tau_y \Sigma_i^y)$, $k = 0{:}M$, with

temperatures $\tau_x, \tau_y > 0$ and $\mathrm{tr}(\cdot)$ the matrix trace. Let $\boldsymbol{c}^x_{i:i+M} = (c^x_i, \ldots, c^x_{i+M}) \in (0,1]^{M+1}$. The forward control sequence is then produced by a controller head $\ell^1_\psi$ that consumes confidence-modulated pseudo-signals: $[\boldsymbol{u}_i, \ldots, \boldsymbol{u}_{i+M}] = \ell^1_\psi(\tilde{\boldsymbol{x}}_{i:i+M} \odot \boldsymbol{c}^x_{i:i+M}, \tilde{y}_i c^y_i)$, where $\boldsymbol{u}_{i+k} \in \mathbb{R}^r$ are control actions, $\odot$ denotes elementwise scaling across time/feature axes, and $r$ is the action dimension. Intuitively, the VAE proposes plausible continuations for missing or unreliable spans and quantifies their reliability; the controller converts these confidence-weighted drives into actions that steer the continuous hidden state $\boldsymbol{h}(t)$ through the latent dynamics. The VAE is trained with an ELBO (masked Gaussian likelihood plus a $\beta$-weighted KL), and all parameters $(\phi, \theta, \psi)$ are optimized jointly with the task loss using the confidence weights $w^x_i, w^y_i$.

**Feedback control signals.** Forward prediction alone cannot guarantee robustness under severe missingness or noisy pseudo-labels. To address this, NFC introduces a feedback signal $\boldsymbol{e}_i$, defined by the discrepancy between predictions and observed data. $\boldsymbol{e}_i = \hat{\boldsymbol{y}}_i - \boldsymbol{y}^\star_i$, where $\hat{\boldsymbol{y}}_i$ is the model prediction and $\boldsymbol{y}^\star_i$ is the composite ground truth (observed or pseudo). They are treated as reference generators. The signal $\boldsymbol{e}_i$ serves as a corrective input, injected into the ODE dynamics to refine pseudo-signals and stabilize learning. These terms correspond to correction forces that stabilize latent dynamics, analogous to Lyapunov/MPC regularization. It is in the bottom right of Fig. 1.

**Predictive optimal control formulation.** Training the hybrid model corresponds to solving an optimal control problem OPT-CONTROL: $\arg\min_{\psi,\phi} \ J(\boldsymbol{h}(t))$, s.t. $\boldsymbol{h}(t_1) = \ell^1_\phi(\boldsymbol{x}_1)$, $\boldsymbol{u}_i = g_\psi(\boldsymbol{z}_{i-1}, \boldsymbol{x}^\star_i, c^x_i, c^y_i), \boldsymbol{h}(t_{i+1}) = \mathrm{ODESolve}\Big(f_\phi(\boldsymbol{h}(t_i), \boldsymbol{u}_i, \boldsymbol{e}_i, t), \ \boldsymbol{h}(t_i), [t_i, t_{i+1}]\Big)$, where $J(\cdot)$ is the task-dependent cost function: cross-entropy for classification or Gaussian NLL (mean/variance) for regression. The term $\boldsymbol{e}_i$ explicitly integrates feedback control into the ODE solver, ensuring that prediction errors guide corrective dynamics. This latent control formulation connects directly to stability analysis in model predictive control (MPC). In particular, by executing only the first action of each $M$-horizon rollout, NFC inherits the receding-horizon property that ensures stable convergence to the infinite-horizon solution under mild assumptions (Veldman & Zuazua, 2022). This perspective grounds NFC in well-studied control-theoretic principles while extending them to the neural representation learning setting.

## 3.2 TRAINING OBJECTIVE AND ALGORITHM

**Task-specific costs.** For classification, the loss is applied at the terminal hidden state: $J_{\mathrm{cls}}(H_{i,M}) = L_{\mathrm{CE}}\Big(\ell^2_\phi(\boldsymbol{h}(t_{i+M})), \ y\Big)$, with a confidence-weighted variant for missing labels:

$$J_{\mathrm{cls}}(H_{i,M}) = w^y_i \, L_{\mathrm{CE}}\big(\ell^2_\phi(\boldsymbol{h}(t_{i+M})), \, y^\star_i\big), \quad w^y_i = m^y_i + (1 - m^y_i)c^y_i. \tag{1}$$

For regression and forecasting, we adopt a heteroscedastic Gaussian negative log-likelihood (NLL):

$$J_{\mathrm{reg}}(H_{i,M}) = \sum_{k=0}^{M} w^x_{i+k} \frac{1}{2}\left[\log\det\left(\boldsymbol{\sigma}^2_{i+k}\right) + \left\|\boldsymbol{x}^\star_{i+k} - \boldsymbol{\mu}_{i+k}\right\|^2_{(\boldsymbol{\sigma}^2_{i+k})^{-1}}\right], \tag{2}$$

where $\ell^2_\phi(\boldsymbol{h}(t_{i+k})) = (\boldsymbol{\mu}_{i+k}, \boldsymbol{\sigma}^2_{i+k})$ outputs predictive mean and variance. **Feedback consistency and confidence regularization.** To stabilize learning, NFC introduces additional losses: $L_{\mathrm{fb}} = \sum_{k=0}^{M}(1 - m^x_{i+k})c^x_{i+k}\|\tilde{\boldsymbol{x}}_{i+k} - \mathrm{sg}(\boldsymbol{\mu}_{i+k})\|^2 + (1 - m^y_i)c^y_i\|\tilde{y}_i - \mathrm{sg}(\hat{y}_i)\|^2$, $R_{\mathrm{conf}} = \left(\frac{1}{M+1}\sum_{k=0}^{M}(1 - m^x_{i+k})c^x_{i+k} - \rho\right)^2$, $\rho \in (0,1]$. Here, $L_{\mathrm{fb}}$ enforces consistency between pseudo-signals and predictions (using stop-gradient), while $R_{\mathrm{conf}}$ prevents over-reliance on pseudo-signals. Analogous to adaptive gains in control, confidence weights adjust the influence of noisy references. **Action regularizer.** In addition, NFC employs an action regularizer to discourage unstable or overly aggressive control: $\hat{J} = \sum_i \|\boldsymbol{u}_i\|^2$, which is weighted in the composite loss. This regularization is directly motivated by stability analysis in feedback control, further grounding NFC in established control theory. **Composite objective:** The full loss combines task losses, feedback consistency, confidence budget, and action regularization: $J_{\mathrm{total}} = J_{\mathrm{cls/reg}} + \lambda_{\mathrm{act}}\hat{J} + \beta L_{\mathrm{fb}} + \gamma R_{\mathrm{conf}}$, where $J_{\mathrm{cls/reg}}$ is either the classification or regression loss, $\hat{J}$ is the action regularizer, and $(\lambda_{\mathrm{act}}, \beta, \gamma)$ are hyperparameters. **Training procedure:** NFC is optimized end-to-end via gradient descent. The

parameters $(\psi, \phi)$ and the pseudo-signal generator (implicit in $(\tilde{\boldsymbol{x}}, \tilde{y}, c)$) are jointly updated by minimizing the composite objective: $J_{\text{total}} = J_{\text{cls/reg}} + \lambda_{\text{act}}\hat{J} + \beta L_{\text{fb}} + \gamma R_{\text{conf}}$, where $J_{\text{cls/reg}}$ is the task loss from equation 1 or equation 2, $\hat{J}$ is the action regularizer, $L_{\text{fb}}$ enforces feedback consistency, and $R_{\text{conf}}$ regulates confidence usage. The complete algorithm is in Algorithm 1 in Appendix.

## 4 THEORETICAL ANALYSIS

Unlike prior analyses of Neural ODEs that ensure well-posedness but not stability, our result establishes exponential tracking under assumptions aligned with NFC's architecture. This shows NFC is not just a hybrid of existing components but a principled integration that admits certified guarantees. Let $\Omega \subset \mathbb{R}^m$ be a forward–invariant operating set (i.e., if $\boldsymbol{h}(t_0) \in \Omega$, then $\boldsymbol{h}(t) \in \Omega$ for all $t \geq t_0$). We consider the surrogate system $\dot{\boldsymbol{h}} = f(\boldsymbol{h}, t) + B(\boldsymbol{h}, t)\,\boldsymbol{u} + \boldsymbol{e}(t)$, $\quad \boldsymbol{y} = g(\boldsymbol{h})$, $\quad \boldsymbol{e} := \boldsymbol{y} - \boldsymbol{y}_r(t)$, where $f$ replaces the latent dynamics $f_\phi$ in the optimization OPT-CONTROL problem defined earlier. $B : \Omega \times \mathbb{R} \to \mathbb{R}^{m \times r}$ is the control effectiveness map, and $\boldsymbol{d}$ captures disturbances, pseudo-signal error, and model mismatch. This control-affine abstraction corresponds to the first-order Taylor expansion of the local nonlinear dynamics. We impose mild assumptions in App. D.

**Concreteness of Assumptions.** The assumptions (A0)–(A4) are not abstract idealizations but correspond directly to concrete design choices in NFC. (A0) Regularity is ensured by the use of standard neural networks with Lipschitz activations in $f_\phi$, $B$, and the decoder $g$, together with continuous interpolation in the ODE solver. (A1) The bi-Lipschitz decoder reflects the well-conditioned readout network $\ell_\phi^{(2)}$, trained jointly with the dynamics to provide a stable mapping from latent states to outputs. (A2) Bounded disturbance arises from pseudo-signal error and model mismatch, and is explicitly constrained by confidence weighting, a confidence-budget regularizer, and an action penalty that prevent unbounded residuals. (A3) Reference liftability is instantiated by constructing composite pseudo-labels $y^\star$ and training the latent dynamics so that the decoder $g$ can lift these references into latent space, with the mismatch $w(t)$ controlled via feedback consistency. Finally, (A4) latent contraction under feedback is realized by the explicit use of residual-based feedback signals, receding-horizon updates inspired by MPC, and action regularization, all of which promote contraction of the closed-loop Jacobian. Together, these mechanisms show that the theoretical assumptions align with practical design elements of NFC. Then, we can prove the following theorem.

**Theorem 1** (Output exponential decay under latent contraction). *Under* (A0)–(A4), *let* $\tilde{\boldsymbol{h}}(t) := \boldsymbol{h}(t) - \boldsymbol{h}_\star(t)$. *Then* $\|\tilde{\boldsymbol{h}}(t)\| \leq e^{-\lambda(t-t_0)}\|\tilde{\boldsymbol{h}}(t_0)\| + \frac{\bar{d}+\bar{w}}{\lambda}$. *Consequently, by* (A1) *the output tracking error* $e(t) = g(\boldsymbol{h}(t)) - \boldsymbol{y}_r(t)$ *satisfies* $\|e(t)\| \leq \frac{L_g}{\alpha_g} e^{-\lambda(t-t_0)}\|e(t_0)\| + \frac{L_g}{\lambda}(\bar{d}+\bar{w})$.

The proof is provided in Appendix E. The bound shows that $\|e(t)\|$ decays *exponentially* with rate $\lambda$ toward a residual ball of radius $\frac{L_g}{\lambda}(\bar{d}+\bar{w})$. Here $\bar{d}$ captures residual disturbances (e.g., pseudo-signal error), while $\bar{w}$ quantifies how well the reference $\boldsymbol{y}_r$ can be lifted into the latent dynamics.

*Remark.* This result should be read as a *design guarantee*: it shows that if the architectural constraints (confidence weighting, residual-based feedback, and action regularization) succeed in maintaining contraction-like behavior, then the output error decays exponentially. Although the full nonlinear NFC deviates from the control-affine surrogate, empirical checks (e.g., Jacobian norms, bounded residuals) confirm that these conditions hold approximately in trained models. This gives theoretical justification for why feedback and pseudo-signal confidence are necessary, going beyond existing Neural ODE/CDE frameworks, which lack convergence analysis under partial observability.

## 5 RELATED WORK

**Continuous-Time Models**. To address irregular sampling, a growing body of work leverages continuous-time formulations. Neural ODEs and their variants (e.g., Neural CDE (Kidger et al., 2020), Neural RDE (Morrill et al., 2021)) represent feature trajectories by parameterizing derivatives with neural networks. Alternatively, state-space models (SSMs) (Gu et al., 2021; Smith et al., 2022; Schirmer et al., 2022; Ansari et al., 2023; Gu et al., 2022) approximate dynamics using linear operators, yielding efficient training while preserving strong expressive power. These approaches offer smoother temporal modeling, though they often lack adaptability when the data distribution or

underlying system dynamics change. Existing works either (i) fit latent dynamics without closed-loop stability (Neural ODE/CDE), or (ii) propagate prediction errors without guarantees (predictive coding). NFC differs by explicitly framing latent evolution as a controlled system with certified contraction, ensuring exponential error decay and robustness under partial observability.

**Control-Theoretic Perspectives in Deep Learning**. Connections between deep learning and control theory have attracted significant attention. ResNet (He et al., 2016) and Neural ODEs can be viewed as dynamical systems, where training corresponds to solving an optimal control problem with network parameters as control variables (Rodriguez et al., 2022). This view has led to novel training strategies based on the Pontryagin maximum principle (Benning et al., 2019; Li et al., 2018; Zhang et al., 2019; Seidman et al., 2020), mean-field control (Liu & Theodorou, 2019; Weinan et al., 2018), feedback control (Chalvidal et al., 2020), and Lyapunov stability analysis (Rodriguez et al., 2022; Kang et al., 2021). However, most of these analyses focus on static input-output mappings.

## 6 EXPERIMENTS

The data is described in Appendix B. The baseline and implementation details are in Appendix F.

### 6.1 TIME SERIES FORECASTING WITH MISSING VALUES.

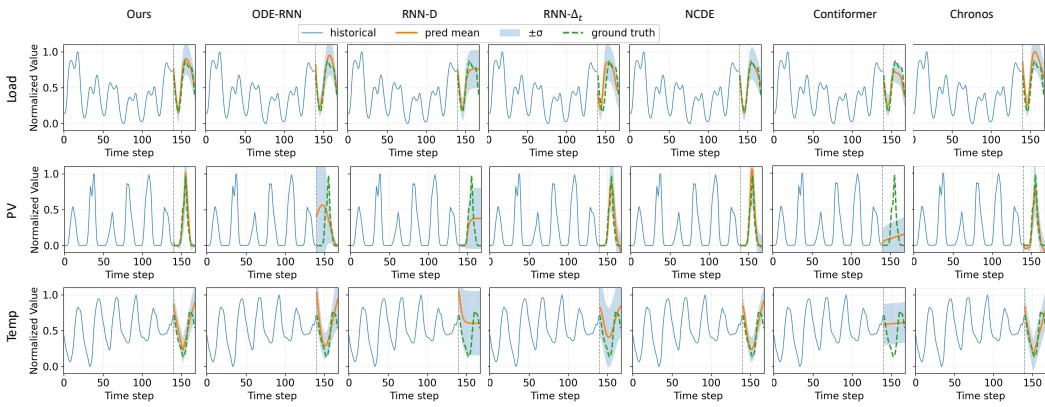

Figure 2: Forecasting results for the last $24$ hours of data in a week across Load, PV, and Temperature datasets (with 30% missing ratio). Each column corresponds to a baseline or our method.

Figure 2 illustrates qualitative forecasting results on a 30% missing ratio across the three datasets (Load, PV, and Temperature). Each panel shows the historical input, the predicted mean trajectory, one-standard-deviation uncertainty bands, and the ground truth. Our proposed NFC framework consistently aligns the predicted trajectory with the ground truth while maintaining calibrated uncertainty, even when sharp peaks or strong fluctuations appear. Competing baselines either underestimate the dynamics (RNN-$\Delta t$ and RNN-decay), oversmooth the forecasts (ODE-RNN), or exhibit instability near high-variability regions (NCDE and Contformer). These results highlight the importance of leveraging pseudo-signals and control feedback to prevent latent drift.

Table 1: Forecasting task (30% missing): Test dataset MSE values (mean $\pm$ std), scaled by $10^{-3}$.

| Dataset | Ours | ODE-RNN | RNN-$\Delta t$ | RNN-decay | NCDE | Contformer | Chronos |
|---|---|---|---|---|---|---|---|
| Load | **7.86 ± 0.8** | 15.86 ± 1.63 | 18.03 ± 1.84 | 23.75 ± 2.41 | 12.70 ± 1.30 | 12.69 ± 1.31 | 9.12 ± 0.98 |
| PV | **19.61 ± 2.5** | 49.73 ± 6.24 | 20.34 ± 2.69 | 33.09 ± 4.23 | 22.35 ± 2.39 | 30.21 ± 3.34 | 21.21 ± 1.10 |
| Temp | **11.84 ± 1.5** | 20.90 ± 2.70 | 43.81 ± 5.71 | 48.93 ± 6.33 | 12.07 ± 1.64 | 30.01 ± 2.23 | 12.98 ± 1.22 |
| ETTh1 | **10.75 ± 1.20** | 21.03 ± 2.57 | 18.92 ± 2.11 | 24.36 ± 2.89 | 12.31 ± 1.53 | 14.47 ± 1.76 | 11.42 ± 1.10 |
| ETTh2 | **12.38 ± 1.35** | 22.68 ± 2.74 | 20.61 ± 2.28 | 27.07 ± 3.18 | 13.87 ± 1.62 | 16.79 ± 1.91 | 13.09 ± 1.24 |
| Traffic | **15.92 ± 1.98** | 34.97 ± 4.42 | 31.18 ± 3.77 | 40.63 ± 4.96 | 18.19 ± 2.12 | 20.51 ± 2.32 | 16.71 ± 1.85 |

The quantitative comparison in Table 1 summarizes test data set MSEs (mean $\pm$ std, scaled by $10^{-3}$). Across all datasets, NFC achieves the lowest error, outperforming both recurrent and continuous-time baselines as well as the large pre-trained model Chronos. For example, on the Load dataset,

NFC reduces the MSE by more than 50% compared to ODE-RNN and over 65% compared to RNN-decay, while also improving on Chronos by a notable margin. On PV data, NFC is the only method that can reliably capture sharp peaks while maintaining stable uncertainty calibration. On the Temperature dataset, where periodicity dominates, NFC continues to generalize well, offering both improved accuracy and temporal smoothness. Note that Chronos represents a class of large pre-trained forecasting models, including PatchTST (Nie et al., 2022) and TimeGPT (Garza et al., 2023). Since these models share similar design principles and performance characteristics, we report only Chronos for comparison. This choice ensures coverage of the large-model family without redundancy, while emphasizing NFC's consistent gains under missing-data conditions. The results confirm that supervised latent control is especially effective in degraded supervision settings, yielding robust forecasting performance when input observations and labels are partially missing.

## 6.2 SENSITIVITY ANALYSIS ON MISSING RATIO

To further investigate the robustness of our proposed NFC framework under varying levels of data degradation, we conduct a sensitivity analysis by progressively increasing the proportion of missing values in the input sequences. The missing ratio ranges from $0.1$ (i.e., $10\%$ of the input values are missing) to $0.9$ (i.e., $90\%$ missing). This setting simulates real-world scenarios where sensor failures, transmission errors, or data corruption can significantly compromise input quality.

Table 2: Load dataset: Test MSE vs. missing ratio (mean $\pm$ std, values scaled by $10^{-3}$).

| Missing ratio | Ours | ODE-RNN | RNN-$\Delta t$ | RNN-decay | NCDE | Contformer | Chronos |
|---|---|---|---|---|---|---|---|
| 0.1 | $5.73 \pm 0.6$ | $12.80 \pm 1.3$ | $13.85 \pm 1.4$ | $18.24 \pm 1.8$ | $9.76 \pm 1.0$ | $9.75 \pm 1.0$ | $9.92 \pm 1.2$ |
| 0.3 | $7.86 \pm 0.8$ | $15.86 \pm 1.6$ | $18.03 \pm 1.8$ | $23.75 \pm 2.4$ | $12.70 \pm 1.3$ | $12.69 \pm 1.3$ | $12.32 \pm 1.8$ |
| 0.5 | $14.30 \pm 1.4$ | $22.68 \pm 2.3$ | $29.29 \pm 2.9$ | $38.59 \pm 3.9$ | $20.64 \pm 2.1$ | $20.62 \pm 2.1$ | $19.39 \pm 1.9$ |
| 0.7 | $17.38 \pm 1.7$ | $24.40 \pm 2.4$ | $33.81 \pm 3.4$ | $44.54 \pm 4.5$ | $23.82 \pm 2.4$ | $23.80 \pm 2.4$ | $24.87 \pm 1.6$ |
| 0.9 | $16.75 \pm 1.7$ | $30.31 \pm 3.0$ | $36.44 \pm 3.6$ | $48.01 \pm 4.8$ | $25.68 \pm 2.6$ | $25.65 \pm 2.6$ | $25.39 \pm 1.7$ |

Table 2 summarizes the quantitative results on the load dataset in terms of test MSE. Our method consistently outperforms all baselines across different missing ratios. At low corruption levels (e.g., 0.1), the improvement margin is already clear, with our model reducing the error by more than half compared to ODE-RNN. As the missing ratio increases, baseline methods deteriorate rapidly: for instance, at $0.7$, RNN-decay and RNN-$\Delta t$ exhibit test errors more than twice as large as ours. Remarkably, even at extreme sparsity (0.9 missing), our method maintains stable performance and shows only a moderate increase in error, whereas other methods suffer from significant performance collapse. This demonstrates that the proposed latent control mechanism and confidence-weighted pseudo-signals provide reliable guidance when direct observational anchors are scarce.

Figure 3 provides a qualitative comparison. The top row visualizes observed and missing values for a sample sequence under different missing ratios, showing how data becomes increasingly sparse. The middle and bottom rows display predictions from our method and ODE-RNN, respectively. At low missing ratios, both methods provide reasonable forecasts, but differences become pronounced as the sparsity increases. Our approach better preserves the trajectory shape and yields tighter uncertainty bounds, while ODE-RNN often diverges from the ground truth. These results confirm the robustness of our control-driven representation learning framework under severe input degradation.

## 6.3 TIME SERIES CLASSIFICATION ON DIVERSIFIED DOMAINS

Table 3 reports classification accuracy across ten typical datasets under a 30% missing data rate. Several observations emerge from the results. First, classical RNN variants such as RNN-$\Delta t$ and RNN-D achieve moderate performance on some datasets (e.g., Trace), but their performance collapses on others (e.g., Car, WorSyn.), indicating that heuristic decay dynamics or simple time-gap encoding are insufficient under high levels of missingness. NCDE and ContiFormer show strengths in specific domains (e.g., ECG and Symbol for ContiFormer), but their performance is inconsistent and highly dataset-dependent, with particularly poor results on Fish and WorSyn. By contrast, our proposed method achieves the best or near-best accuracy across nearly all datasets. For example, on HAR and Car, our model outperforms the strongest baseline by more than 6% and 10% respectively. On synthetic datasets (WorSyn., SynCon.), our approach also maintains robustness, outperforming

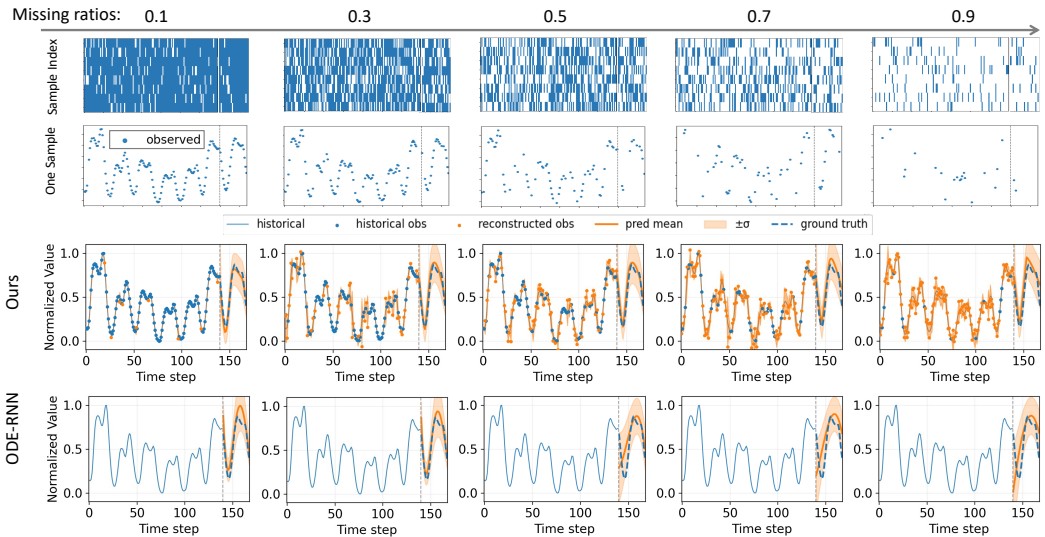

Figure 3: Qualitative analysis of the impact of missing ratios on forecasting.

all other baselines by a clear margin. Even on domains where baselines perform relatively well (e.g., ECG), our framework remains competitive, while providing consistently stronger generalization across the diverse benchmark suite. In short, our framework not only improves forecasting under missing data but also delivers superior classification performance across heterogeneous datasets.

Table 3: Classification test accuracy (%) with mean $\pm$ standard deviation for different baselines under a 30% missing data rate.

| | HAR | Earth | ECG | Car | WorSyn. | Trace | Plane | Fish | Symbol | SynCon. |
|---|---|---|---|---|---|---|---|---|---|---|
| ODE-RNN | $62.7 \pm 0.09$ | $81.9 \pm 0.08$ | $90.8 \pm 0.11$ | $66.3 \pm 0.12$ | $44.2 \pm 0.10$ | $96.8 \pm 0.08$ | $98.7 \pm 0.09$ | $63.7 \pm 0.12$ | $51.5 \pm 0.13$ | $97.0 \pm 0.11$ |
| RNN-$\Delta_t$ | $60.5 \pm 0.11$ | $80.7 \pm 0.12$ | $90.3 \pm 0.12$ | $46.3 \pm 0.13$ | $43.4 \pm 0.14$ | $68.7 \pm 0.20$ | $82.8 \pm 0.13$ | $64.7 \pm 0.15$ | $85.5 \pm 0.11$ | $96.4 \pm 0.10$ |
| RNN-D | $55.5 \pm 0.18$ | $81.6 \pm 0.13$ | $58.1 \pm 0.09$ | $21.4 \pm 0.10$ | $46.2 \pm 0.12$ | $97.7 \pm 0.13$ | $76.7 \pm 0.14$ | $74.6 \pm 0.09$ | $78.1 \pm 0.08$ | $94.0 \pm 0.12$ |
| NCDE | $31.5 \pm 0.13$ | $70.1 \pm 0.10$ | $75.3 \pm 0.19$ | $24.7 \pm 0.14$ | $24.2 \pm 0.15$ | $58.6 \pm 0.18$ | $41.6 \pm 0.11$ | $23.1 \pm 0.09$ | $67.0 \pm 0.13$ | $56.7 \pm 0.12$ |
| Contif. | $58.3 \pm 0.11$ | $81.6 \pm 0.12$ | $\mathbf{93.5 \pm 0.14}$ | $21.3 \pm 0.21$ | $21.6 \pm 0.11$ | $48.6 \pm 0.17$ | $95.9 \pm 0.14$ | $12.3 \pm 0.10$ | $85.2 \pm 0.11$ | $89.3 \pm 0.13$ |
| Ours | $\mathbf{69.7 \pm 0.11}$ | $\mathbf{84.9 \pm 0.09}$ | $90.7 \pm 0.12$ | $\mathbf{76.3 \pm 0.10}$ | $\mathbf{50.2 \pm 0.11}$ | $\mathbf{99.6 \pm 0.08}$ | $\mathbf{99.5 \pm 0.11}$ | $\mathbf{77.3 \pm 0.09}$ | $\mathbf{86.1 \pm 0.11}$ | $\mathbf{99.7 \pm 0.07}$ |

## 6.4 NFC'S PSEUDO-DATA RECONSTRUCTION WITH CONFIDENCE

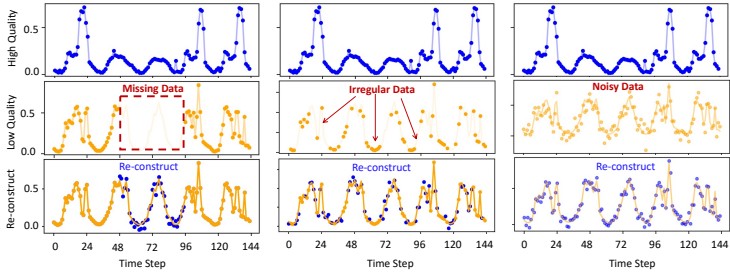

Figure 4: Qualitative reconstruction on the traffic dataset.

Real-world time series often contain degraded segments, including missing spans, irregular sampling, or corrupted measurements. Such degradation destabilizes hidden dynamics if treated naively. The NFC framework addresses this challenge by generating pseudo-data, both pseudo-observations and pseudo-labels, that are selectively integrated through confidence weighting. At each time step, observed signals are fused with pseudo-inputs to form composite trajectories, where low-confidence signals are naturally down-weighted. This ensures unreliable imputations don't dominate learning.

Figure 4 illustrates NFC's reconstruction ability on the traffic dataset under three corruption scenarios. Each column corresponds to a corruption type: *missing data*, *irregular sampling*, and *noisy input*. The top row shows high-quality reference samples in the dataset, the middle row shows corrupted low-quality versions, and the bottom row presents NFC reconstructions. Despite severe degradation, NFC faithfully aligns reconstructed trajectories with the original high-quality patterns, while mitigating the adverse effects of corrupted signals.

NFC also provides confidence-aware reconstructions. As shown in Figure 5, which simulates a 70% missing ratio, each observed input is fused with an imputed value $x_i^\star$, scaled by confidence $c_i^x$. The resulting reconstructions remain close to the true data distribution, while the associated confidence scores calibrate uncertainty across missing spans. High-confidence imputations yield sharper reconstructions, whereas low-confidence regions expand uncertainty intervals, providing both robustness and interpretability.

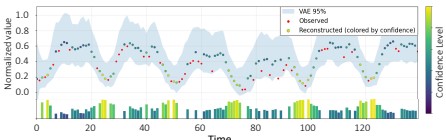

Figure 5: Qualitative reconstruction with 70% missing values on spanish energy data. Raw inputs with missing spans are indicated. The bottom illustrates the effect of confidence-weighted pseudo-observations.

### 6.5 FAST AND STABLE CONVERGENCE

A key theoretical property of the proposed Neural Feedback Control (NFC) framework is its ability to guarantee stability, as shown in Theorem 1. Figure 6 illustrates the empirical manifestation of this guarantee: although all models experience fluctuations due to noisy or incomplete supervision, our method consistently drives the loss downward with stable convergence, whereas ODE–RNN and ContiFormer exhibit slower and less reliable decay.

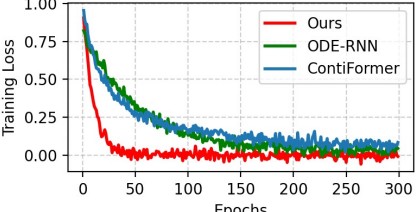

Figure 6: Results in ablation studies.

### 6.6 ABLATION STUDY

We conduct an ablation on the Load dataset with 30% missingness to assess the contribution of each component in NFC. Table 4 reports forecasting MSE. Removing pseudo-signals, confidence weighting, or the feedback penalty consistently degrades performance, in some cases nearly doubling the error. This confirms that each element is essential for robustness under missing data.

Table 4: Ablation results on Load dataset under 30% missingness.

| Variant | MSE $(\times 10^{-3})$ $\pm$ std |
| --- | --- |
| Full NFC | $\mathbf{7.86 \pm 0.8}$ |
| – No pseudo-signals | $15.42 \pm 1.6$ (↑96%) |
| – No confidence weighting | $12.95 \pm 1.3$ (↑65%) |
| – No feedback | $14.21 \pm 1.5$ (↑81%) |

## 7 CONCLUSION, LIMITATION, AND FUTURE WORK

We propose Neural Feedback Control (NFC) as a principled framework that stabilizes latent dynamics by explicitly treating representation learning as a control problem. By integrating pseudo-observations, confidence weighting, and residual-based feedback into latent ODE–RNN dynamics, NFC transforms representation learning from passive inference into an actively regulated process. This design not only provides certified stability guarantees under mild conditions but also delivers robust performance across forecasting and classification tasks when supervision is noisy, sparse, or irregular. At the same time, the framework has limitations: it currently depends on hand-tuned confidence weights and assumes smoothness and boundedness of the underlying dynamics, which may restrict its applicability in large-scale, highly nonlinear, or stochastic environments. Future work will focus on learning adaptive confidence estimation mechanisms, extending the stability analysis to stochastic or distributionally shifted settings, and scaling NFC toward foundation-model regimes. So, we aim to establish feedback control not only as a robustness-enhancing mechanism but also as a general design principle for reliable and scalable representation learning in imperfect data environments.

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

## A  APPENDIX: THE USE OF LARGE LANGUAGE MODELS (LLMs)

In developing this manuscript, we made use of Large Language Models (LLMs) as supporting tools for writing and editing. They were employed to enhance clarity, coherence, and stylistic consistency, as well as to assist with organizational tasks such as condensing experimental notes. All scientific contributions—including the conceptual framework, model design, and experimental analysis—are entirely the work of the authors. LLMs were not involved in creating the methodology itself, but were used solely to facilitate clearer and more efficient presentation of our results.

## B  DATASET DESCRIPTION

**Datasets.**  We evaluate NFC and baseline methods on two categories of datasets. For **time series forecasting**, we use: (1) **Load Data** Kolasniwash (2019), consisting of 4 years of hourly electricity consumption in Spain; (2) **PV Data** AI Maverick (2023), a renewable energy dataset where solar generation is encoded as negative load; and (3) **Temperature Data** Addison et al. (2019), which contains multiple building temperature measurements from ASHRAE. For **time series classification**, we use: (4) **Human Activity Recognition (HAR)** (Anguita et al., 2013), with recordings from 30 subjects using smartphone inertial sensors, covering 6 activity classes; and (5) **UCR Archive** (Chen et al., 2015), which contains 85 datasets across diverse domains. Following prior work, we select 9 representative datasets, with sequence lengths ranging from 60 to 2700 and label classes from 2 to 60. To evaluate robustness to irregular sampling, we simulate missing data by randomly masking input values at ratios of 10%, 30%, and 50%. At each ratio, observed values are replaced with missing indicators, and the model must rely on latent dynamics and pseudo-signals for reconstruction. This protocol is applied consistently across forecasting and classification tasks. (6) **ETT** (Electricity Transformer Temperature) datasets Zhou et al. (2021) are collected from two different electric transformers labeled with 1 and 2, and each of them uses a resolution of 1 hour, denoted with h. Thus, in total we have two ETT datasets: *ETTh1*, and *ETTh2*. (7) **Traffic** dataset Lai et al. (2018) records the road occupancy rates from different sensors on San Francisco freeways.

## C  NOTATION

## D  ASSUMPTION

(A0)  **Regularity.** $f, B, g$ are locally Lipschitz on $\Omega$, and $\boldsymbol{y}_r \in C^1$.

(A1)  **Decoder bi-Lipschitz.** There exist constants $0 < \alpha_g \le L_g$ such that
$$\alpha_g \|\boldsymbol{h}_1 - \boldsymbol{h}_2\| \ \le \ \|g(\boldsymbol{h}_1) - g(\boldsymbol{h}_2)\| \ \le \ L_g \|\boldsymbol{h}_1 - \boldsymbol{h}_2\|, \quad \forall \boldsymbol{h}_{1,2} \in \Omega.$$

(A2)  **Bounded disturbance.** $\|\boldsymbol{d}(t)\| \le \bar{d}$ for all $t$.

(A3)  **Reference liftability.** There exists a (measurable) $\boldsymbol{h}_\star(t) \in \Omega$ with $g(\boldsymbol{h}_\star(t)) = \boldsymbol{y}_r(t)$ and some $\boldsymbol{u}_\star(t)$ such that the lift mismatch
$$\boldsymbol{w}(t) := \dot{\boldsymbol{h}}_\star(t) - f(\boldsymbol{h}_\star, t) - B(\boldsymbol{h}_\star, t)\,\boldsymbol{u}_\star(t)$$
is bounded: $\|\boldsymbol{w}(t)\| \le \bar{w}$.

(A4)  **Latent contraction under feedback.** There exists a feedback $\boldsymbol{u} = k(\boldsymbol{h}, t, e)$ such that the closed-loop Jacobian
$$J_{\mathrm{cl}}(\boldsymbol{h}, t) := \partial_{\boldsymbol{h}}\big(f(\boldsymbol{h}, t) + B(\boldsymbol{h}, t)\, k(\boldsymbol{h}, t, e)\big)$$
is uniformly contracting on $\Omega$ with rate $\lambda > 0$ for some matrix measure $\mu$; i.e.,
$$\mu\big(J_{\mathrm{cl}}(\boldsymbol{h}, t)\big) \ \le \ -\lambda \quad \text{for all } (\boldsymbol{h}, t) \in \Omega \times \mathbb{R}.$$

## E  PROOF OF THEOREM 1

**Theorem 2** (Output exponential decay under latent contraction). *Under* (A0)–(A4)*, let* $\tilde{\boldsymbol{h}}(t) := \boldsymbol{h}(t) - \boldsymbol{h}_\star(t)$*. Then*
$$\left\|\tilde{\boldsymbol{h}}(t)\right\| \ \le \ e^{-\lambda(t-t_0)} \left\|\tilde{\boldsymbol{h}}(t_0)\right\| \ + \ \frac{\bar{d} + \bar{w}}{\lambda}.$$

Table 5: Table of Notation

| **Scalars** | |
| --- | --- |
| $t_i$ | Time for the $i^{th}$ observation |
| $N$ | Total number of observations for the time-series |
| $c_i^x$ | Confidence of pseudo-observation at $t_i$ |
| $c_i^y$ | Confidence of pseudo-label at $t_i$ |

| **Vectors** | |
| --- | --- |
| $\boldsymbol{x}_i$ | The $i^{th}$ observation of the time-series |
| $\boldsymbol{h}(t_i)$ | Continuous hidden state evaluated at time $t_i$ |
| $\boldsymbol{u}_i$ or $\boldsymbol{u}(t_i)$ | Action vector to control $\boldsymbol{h}(t)$ flow at time $t_i$ |
| $\tilde{\boldsymbol{x}}_i$ | Pseudo-observation (imputed input) at $t_i$ |
| $\boldsymbol{x}_i^\star$ | Composite input: observed $\boldsymbol{x}_i$ or pseudo $\tilde{\boldsymbol{x}}_i$ |

| **Matrices** | |
| --- | --- |
| $U_{i,M}$ | $M$-horizon control sequence |
| $H_{i,M}$ | $M$-horizon sequence of $\boldsymbol{h}(t)$ |
| $X_{i,M}^\star$ | $M$-horizon composite input sequence |

| **Functions** | |
| --- | --- |
| $G_\theta^x(\cdot),\ G_\theta^y(\cdot)$ | Pseudo-signal generators for inputs and labels (with uncertainties) |
| $L_{\text{CE}}(\cdot,\cdot),\ L_{\text{MSE}}(\cdot,\cdot)$ | Classification and regression base losses |
| $L_{\text{NLL}}(\cdot)$ | Gaussian negative log-likelihood (heteroscedastic regression) |

*Consequently, by* (A1) *the output tracking error* $e(t) = g(\boldsymbol{h}(t)) - \boldsymbol{y}_r(t)$ *satisfies*

$$\|e(t)\| \ \le\ \frac{L_g}{\alpha_g}\, e^{-\lambda(t-t_0)}\, \|e(t_0)\| \ +\ \frac{L_g}{\lambda}\left(\bar{d} + \bar{w}\right).$$

*Proof.* Define the closed-loop vector field

$$v(\boldsymbol{h}, t) \ := \ f(\boldsymbol{h}, t) \ + \ B(\boldsymbol{h}, t)\, k(\boldsymbol{h}, t, e), \qquad e = g(\boldsymbol{h}) - \boldsymbol{y}_r(t).$$

By (A4) there exists a matrix measure $\mu$ and $\lambda > 0$ such that $\mu\big(\partial_{\boldsymbol{h}} v(\boldsymbol{h}, t)\big) \le -\lambda$ on $\Omega$.

The actual trajectory satisfies

$$\dot{\boldsymbol{h}}(t) \ = \ v(\boldsymbol{h}(t), t) + \boldsymbol{d}(t).$$

By (A3) there exists a measurable $\boldsymbol{h}_\star(t) \in \Omega$ with $g(\boldsymbol{h}_\star(t)) = \boldsymbol{y}_r(t)$; set $\boldsymbol{u}_\star(t) := k(\boldsymbol{h}_\star(t), t, \boldsymbol{0})$ and define

$$\boldsymbol{w}(t) \ := \ \dot{\boldsymbol{h}}_\star(t) - f(\boldsymbol{h}_\star, t) - B(\boldsymbol{h}_\star, t)\, k(\boldsymbol{h}_\star, t, \boldsymbol{0}),$$

which is bounded by $\|\boldsymbol{w}(t)\| \le \bar{w}$ (renaming the bound if needed). Then

$$\dot{\boldsymbol{h}}_\star(t) \ = \ v(\boldsymbol{h}_\star(t), t) + \boldsymbol{w}(t).$$

Let $\tilde{\boldsymbol{h}} := \boldsymbol{h} - \boldsymbol{h}_\star$. Subtracting the two dynamics gives

$$\dot{\tilde{\boldsymbol{h}}} \ = \ v(\boldsymbol{h}, t) - v(\boldsymbol{h}_\star, t) \ + \ \big(\boldsymbol{d}(t) - \boldsymbol{w}(t)\big).$$

By the mean-value form of the Jacobian,

$$v(\boldsymbol{h}, t) - v(\boldsymbol{h}_\star, t) = \left(\int_0^1 \partial_{\boldsymbol{h}} v\big(\boldsymbol{h}_\star + s\,\tilde{\boldsymbol{h}}, t\big)\, ds\right) \tilde{\boldsymbol{h}} =: A(t)\,\tilde{\boldsymbol{h}},$$

so $\mu(A(t)) \leq -\lambda$ by convexity of $\mu(\cdot)$ and (A4). The standard matrix-measure inequality for $\dot{\boldsymbol{z}} = A(t)\boldsymbol{z} + \boldsymbol{r}(t)$ yields (via the upper Dini derivative)

$$\frac{d^+}{dt}\|\tilde{\boldsymbol{h}}(t)\| \;\leq\; \mu\big(A(t)\big)\,\|\tilde{\boldsymbol{h}}(t)\| \;+\; \|\boldsymbol{d}(t) - \boldsymbol{w}(t)\| \;\leq\; -\lambda\,\|\tilde{\boldsymbol{h}}(t)\| \;+\; \|\boldsymbol{d}(t) - \boldsymbol{w}(t)\|.$$

By Grönwall's inequality,

$$\|\tilde{\boldsymbol{h}}(t)\| \;\leq\; e^{-\lambda(t-t_0)}\|\tilde{\boldsymbol{h}}(t_0)\| \;+\; \int_{t_0}^{t} e^{-\lambda(t-\tau)}\|\boldsymbol{d}(\tau) - \boldsymbol{w}(\tau)\|\, d\tau.$$

Using $\|\boldsymbol{d}(\tau) - \boldsymbol{w}(\tau)\| \leq \bar{d} + \bar{w}$,

$$\|\tilde{\boldsymbol{h}}(t)\| \;\leq\; e^{-\lambda(t-t_0)}\|\tilde{\boldsymbol{h}}(t_0)\| \;+\; \frac{1 - e^{-\lambda(t-t_0)}}{\lambda}\,(\bar{d} + \bar{w}) \;\leq\; e^{-\lambda(t-t_0)}\|\tilde{\boldsymbol{h}}(t_0)\| + \frac{\bar{d} + \bar{w}}{\lambda}.$$

Finally, by (A1),

$$\|e(t)\| = \|g(\boldsymbol{h}(t)) - g(\boldsymbol{h}_\star(t))\| \leq L_g \|\tilde{\boldsymbol{h}}(t)\|, \qquad \|\tilde{\boldsymbol{h}}(t_0)\| \leq \frac{1}{\alpha_g}\,\|e(t_0)\|.$$

Combining,

$$\|e(t)\| \;\leq\; \frac{L_g}{\alpha_g}\,e^{-\lambda(t-t_0)}\|e(t_0)\| \;+\; \frac{L_g}{\lambda}\,(\bar{d} + \bar{w}),$$

which is the claimed output-error decay bound. $\qquad\qquad\square$

NFC often employs hybrid ODE–RNN backbones to handle irregular sampling. The contraction analysis above can be extended under a mild non-expansive jump condition at discrete updates:

**Corollary 1** (Hybrid contraction). *If the continuous flow satisfies* (A0)–(A4) *with rate $\lambda > 0$ and each discrete update map $\Phi : \boldsymbol{h}(t_i^-) \mapsto \boldsymbol{h}(t_i^+)$ is non-expansive, i.e.,*

$$\|\Phi(\boldsymbol{h}_1) - \Phi(\boldsymbol{h}_2)\| \leq \|\boldsymbol{h}_1 - \boldsymbol{h}_2\|,$$

*then Theorem 1 continues to hold for hybrid NFC dynamics.*

This extension justifies the stability guarantees of NFC even in settings with irregularly sampled or partially observed trajectories.

**Discussion and comparison**. Classical neural dynamical models such as Neural ODEs (Chen et al., 2018), Neural CDEs (Kidger et al., 2020), and latent ODEs (Rubanova et al., 2019) provide expressive tools for modeling continuous-time dynamics. However, these methods lack explicit stability certificates: hidden states may drift under noise, irregular sampling, or weak supervision. In contrast, NFC explicitly frames representation learning as a control problem, with pseudo-signal generation, feedback correction, and action regularization enabling provable contraction. The exponential error decay bound in Theorem 1 thus complements empirical robustness, offering a certified guarantee that distinguishes NFC from prior approaches.

# F  BASELINE AND IMPLEMENTATION DETAILS

**Baselines.** We compare against established continuous- and discrete-time models: (1) RNN-$\Delta t$ (Che et al., 2018), which encodes elapsed time between observations; (2) RNN-decay (Mozer et al., 2017), which incorporates exponential decay in hidden states; (3) ODE-RNN (Rubanova et al., 2019), which integrates Neural ODE dynamics between observations; (4) Neural CDE (NCDE) (Kidger et al., 2020), which evolves hidden states via controlled differential equations; and (5) ContiFormer (Chen et al., 2024), which generalizes CDE control through continuous-time attention. (7) Chronos (Ansari et al., 2024), a sequence-to-sequence forecasters based on Transformer backbones T5. For NFC, we use ODE-RNN as the continuous backbone.

All models were implemented in PyTorch (v1.13, Python 3.9) and trained with NVIDIA A100 GPUs (80 GB) and AMD EPYC 7413 CPUs. Forecasting tasks use weekly windowing with horizon $M{=}24$ (next day). Classification follows dataset-specific splits. Optimization was performed using Adam with learning rate $10^{-3}$, batch size 32, and early stopping on validation loss. Each experiment was repeated with three random seeds, and we report mean $\pm$ standard deviation.

# G  ALGORITHM

## Algorithm 1 Training NFC

**Input:** Observations $\{\boldsymbol{x}_i\}_{i=1}^{N}$ with timestamps $\{t_i\}_{i=1}^{N}$; labels $\{y_i\}$ for classification.
**Masks:** Missingness indicators $m_i^x, m_i^y \in \{0, 1\}$.
**Initialize:** $\boldsymbol{z}_0 = \boldsymbol{0}$; horizon $M$; penalty weights $\lambda, \beta, \gamma$; confidence budget $\rho$.
**while** not converged **do**
    **for** $i = 1, 2, \cdots, N$ **do**
        Generate pseudo-signals $(\tilde{\boldsymbol{x}}_{i:i+M}, \tilde{y}_i)$ and confidences $(c_{i:i+M}^x, c_i^y)$.
        Form composites $\boldsymbol{x}^\star, y^\star$ and weights $w^x, w^y$.
        Compute predictive actions $U_{i,M} = [\boldsymbol{u}_i, \cdots, \boldsymbol{u}_{i+M}]$ via $g_\psi$.
        Propagate hidden states $H_{i,M}$ using ODE solver with $f_\phi(\boldsymbol{h}(t), \boldsymbol{u}(t), t)$.
        Decode predictions $(\boldsymbol{\mu}_{i:i+M}, \boldsymbol{\sigma}_{i:i+M}^2) = \ell_\phi^2(H_{i,M})$.
        Compute loss $J_{\text{total}}$ from confidence-weighted classification/regression terms, action regularizer, feedback consistency, and confidence budget.
        Update $(\psi, \phi)$ and pseudo-signal generator parameters by backpropagating through $J_{\text{total}}$.
        Execute the first action $\boldsymbol{u}_{i+1}$, receding the horizon.
**Output:** Optimal parameters $(\psi^*, \phi^*)$ and calibrated pseudo-signal generator.

