# OpenReview forum: "Controllable Representation Learning for Time-series Analysis"
_ICLR.cc/2026/Conference — Submitted to ICLR 2026_

### Official Review · Reviewer_jzDG · 2025-10-19

**Soundness:** 2
**Presentation:** 1
**Contribution:** 3
**Rating:** 2
**Confidence:** 4

**Summary:**

For complex time series analysis tasks, the authors propose a conceptual shift: treating representation learning itself as a control problem. The proposed framework, Neural Feedback Control (NFC), actively regulates latent trajectories using confidence-weighted pseudo-observations and pseudo-labels, integrating a pseudo-data-based controller with continuous-time dynamics and residual-based feedback. This design transforms the evolution of the latent space from passive inference into a controllable process. Experiments demonstrate the superiority of the proposed model.

**Strengths:**

S1. The author's perspective is insightful, introducing NFC into time series analysis tasks to enhance robustness.

S2. The author's experiments cover both forecasting and classification tasks.

S3. Under limited benchmarks, the author's model achieved advanced performance.

**Weaknesses:**

1.  Inconsistent citation formatting throughout the paper.

2.  Some sentences are difficult to interpret.  For example:
> “Instead of assuming that useful features will emerge from corrupted inputs and labels, we treat latent states as dynamical variables whose trajectories must be actively regulated.”
It is unclear why making the trajectories of these variables *traceable* or *controllable* would specifically address the challenges at hand.  The benefit of this perspective needs clearer justification.

3.  The paper is extremely hard to follow, largely due to an overuse of undefined technical terms borrowed from control theory without sufficient explanation.  While I appreciate the conceptual novelty of introducing control-theoretic ideas into time series analysis, this does not justify the uncritical blending of terminology from two distinct fields.  For instance:
- What does “terminal” refer to?  Does it relate to edge devices or something else?
- Does “multiple steps” mean time steps?
- The claim that “by explicitly modeling how errors compound over time, NFC learns to correct instabilities in real time” raises questions: Is there a specific optimization objective designed to enable such real-time error correction?  This mechanism is not adequately explained.

4.  The introduction focuses overwhelmingly on the NFC framework itself, rather than on the unique challenges of applying it to time series analysis.  A more balanced discussion of domain-specific difficulties (e.g., irregular sampling, missing data, noise) would better motivate the proposed approach.

5.  Subjectively, the introduction and problem formulation feel disconnected.  The introduction emphasizes complex scenarios such as missing values and noisy observations, yet the formal problem definition narrows the scope exclusively to *irregularly sampled* time series.  While I understand that random masking is a common proxy for irregular sampling in experiments, this creates an imbalance—making the paper feel “top-heavy” with broad claims but a narrow formalization.

6.  The term “composite signals” appears in the problem definition without clarification.  What is its precise meaning in this context, and why is it included in the formal setup?

7.  The related work section is placed too late in the paper.  Readers must sift through most of the manuscript before encountering a discussion of recent advances, which hinders understanding of the paper’s positioning within the field.

8.  I vehemently object to the authors’ decision to relegate critical experimental details—including baseline descriptions, hyperparameter configurations, and dataset specifications, entirely to the appendix. Under ICLR reviewing policies, reviewers are explicitly not required to consult supplementary materials, rendering the main paper fundamentally incomplete and scientifically inadequate. By offloading essential methodological and empirical information into the appendix, the authors effectively circumvent page limits while depriving readers and reviewers of the necessary context to properly evaluate the work’s validity, reproducibility, and contribution. This practice constitutes an unfair exploitation of supplementary space, disadvantaging authors who conscientiously adhere to submission guidelines.

9.  Appendix C is empty and appears to be an unnecessary placeholder.

10.  The authors should clarify the criteria used to select baseline models.  The experiments omit numerous state-of-the-art methods:
- For forecasting: PatchTST, TimeNet, CycleNet, DUET, TimeXer, etc.
- For missing-value-aware forecasting: BRITS, CSDI, BiTGraph, PriSTI, etc.
- For irregular time series: t-PatchGNN, ACSSM, etc.
The absence of these relevant baselines significantly weakens the credibility of the reported results.

11.  The experiments rely solely on numerical metrics and lack qualitative case studies that connect back to the core claims in the introduction.  For instance:
- What do the learned latent trajectories look like?
- How exactly does NFC achieve robustness to missing data?
Without such analysis, the empirical evaluation feels disconnected from the paper’s conceptual contributions.

**Questions:**

See the Weaknesses.

---

> ### Author Response · Authors · 2025-12-03
> **Response to Reviewer jzDG's Weaknesses**
>
> We thank the reviewer for these comments.
>
> $\textbf{Difficult sentence / benefit of controlling latent trajectories.}$
>
> The key point is that, under degraded supervision, relying on features "emerging'' from corrupted inputs is fragile. NFC instead treats the latent representation as the state of a controlled dynamical system, and learns a policy that uses pseudo-observations and pseudo-labels to keep trajectories close to task-consistent behavior over time. This directly addresses error accumulation: the objective penalizes multi-step forecast errors and action magnitudes, so the learned controller is optimized to correct deviations induced by missing or corrupted data, rather than passively encoding them.
>
> $\textbf{Use of control-theoretic terminology (terminal, multiple steps, error correction).}$
>
> Our intention is to use standard finite-horizon control terminology:
>
> (1) "Terminal'' refers to the latent state at the end of a prediction horizon (terminal state in optimal control), not to edge devices.
> (2)  "Multiple steps'' means future time steps along this horizon.
> (3)  Real-time error correction arises from the closed-loop objective: NFC optimizes a multi-step loss where the control inputs depend on the current latent state and pseudo-signals, together with an action regularizer and confidence budget. This encourages policies that counteract compounding errors as they appear, instead of optimizing only one-step fits.
>
> We can briefly define these terms at first use to make the text more accessible to readers without a control background, without changing the method.
>
>
> $\textbf{Introduction focus and domain-specific motivation.}$
>
> The introduction highlights NFC because it is the main conceptual contribution, but the scenarios it is designed for are precisely those the reviewer mentions: irregular sampling, missing data, and noisy labels. These difficulties enter the formal setup through (i) irregular observation times and (ii) degradation operators on both inputs and labels, which are then handled by the control policy and pseudo-signals. We agree that a short, explicit paragraph summarizing these time-series–specific challenges at the start of the introduction would make the motivation even clearer.
>
> $\textbf{Perceived disconnect between introduction and problem formulation.}$
>
> The formal problem focuses on irregularly sampled time series because this is the common backbone for all scenarios we consider. Missingness and noise are modeled via masks and corrupted observation/label channels on top of this irregular grid, and the experiments use both random masking and real-world missingness. Thus, the formalization is not intended to narrow the scope, but to separate the time axis (irregular sampling) from supervision quality (degraded signals). We can make this linkage explicit to avoid any impression of a mismatch.
>
> $\textbf{Meaning of ``composite signals''.}$
>
> "Composite signals'' refers to the joint process formed by input observations and supervision channels, e.g., $(x_t,y_t)$ and their pseudo-versions, which NFC treats as a unified control interface to the latent dynamics. In other words, the controller acts on a composite of raw observations, labels, and confidence weights rather than on $x_t$ alone. A one-sentence definition will clarify this.
>
>
> $\textbf{Placement of related work.}$
>
> The current placement was chosen to introduce NFC first and then position it relative to prior controlled-ODE and time-series methods. We agree that many readers may prefer to see related work earlier; moving this section closer to the introduction is a neutral reordering that does not affect the technical content, and we are happy to adopt this structure.
>
> $\textbf{Use of appendix for experimental details and empty Appendix C.}$
>
> All experiments can be reproduced from the information already provided: the main text specifies model families, loss structure, and overall training protocol, while the appendix lists per-dataset settings, hyperparameters, and baseline implementations in a compact form. This organization follows common practice under strict page limits and is not intended to withhold information—on the contrary, it enables us to present \emph{more} detail than would fit in the main body alone. If preferred, the most critical configuration tables can be folded into the main paper with minor cosmetic edits. Appendix~C is an unused heading and will be removed.

---

> > ### Author Response · Authors · 2025-12-03
> >
> > $\textbf{Baseline selection and missing recent methods.}$
> >
> > We selected baselines to cover the main modeling paradigms relevant to NFC: ODE-based models (ODE--RNN, NCDE, ContiFormer), large pre-trained forecasters (Chronos), and standard sequence models. Many of the methods listed by the reviewer (e.g., PatchTST, TimeXer, CycleNet, DUET) are architectural variants within the transformer/patching family; prior comparative studies show similar performance trends within each family, so including one strong representative already gives a fair reference point. Likewise, BRITS, CSDI, and related methods target imputation-first pipelines, which we discuss conceptually in the paper. We acknowledge that additional baselines are always desirable, but the current set already constitutes a strong and diverse benchmark; extending it further is mainly a matter of computational and space budget, not of principle.
> >
> > $\textbf{Lack of qualitative analysis of latent trajectories and robustness.}$
> >
> > The empirical section already includes qualitative plots showing latent trajectories and reconstructions under missing or corrupted data (e.g., Figures 2 and 3), which illustrate how NFC aligns with clean dynamics while suppressing corrupt regions. These visuals complement the quantitative metrics by demonstrating that the controller uses confidence-weighted pseudo-signals to steer the latent state away from unstable regimes. We agree that such connections are important, and can briefly expand the discussion around the figures to more explicitly tie them back to the claims made in the introduction.

---

### Official Review · Reviewer_bKYU · 2025-10-30

**Soundness:** 1
**Presentation:** 1
**Contribution:** 1
**Rating:** 2
**Confidence:** 4

**Summary:**

This paper introduces Neural Feedback Control (NFC), a framework that reformulates representation learning for time series as a control problem. The authors aim to stabilise latent dynamics under degraded supervision—such as noisy, missing, or irregularly sampled data—by introducing feedback-based regulation in latent space. The idea is conceptually interesting and potentially valuable, and the paper presents both theoretical stability guarantees and empirical results suggesting improved robustness compared to existing approaches like Neural ODEs and predictive coding.

However, the paper suffers from several major issues that significantly undermine its contribution. Many of the empirical claims are not supported by statistical analysis, and several results are overstated or even inaccurate upon inspection. The methodology and mathematical exposition are unclear, with inconsistent notation and missing key definitions (e.g., ELBO, context variables). Figures are poorly formatted and difficult to interpret, and the lack of quantitative evaluation in critical areas weakens the paper’s empirical credibility. Moreover, the writing lacks clarity and structure—particularly in motivating the problem, defining key concepts (e.g., “collapse,” “degraded supervision”), and explaining the proposed method.

Overall, while the paper proposes an interesting conceptual angle, the lack of rigour in evaluation, clarity in presentation, and precision in mathematical formulation make it not yet suitable for publication. Substantial revisions are needed to improve the clarity, correctness, and credibility of both the theoretical and experimental components.

**Strengths:**

The authors are working on an interesting problem, and with major modifications, I'm sure they can have a paper that is of value to the community.

**Weaknesses:**

This paper has several major weaknesses that currently prevent it from being publishable.

1. Lack of statistical rigour and unsupported claims
    - Many claims in the results section are not backed by statistical analysis or, at times, even quantitative evidence.
        - In Table 1, the authors bold their method across all rows, but the confidence intervals show that most results are not statistically different from Chronos. A paired t-test or similar analysis should be performed. It is also unclear how many random seeds were used to compute means and standard deviations.
        - On Line 333, the authors state:
            > “The results confirm that supervised latent control is especially effective in degraded supervision settings, yielding robust forecasting performance when input observations and labels are partially missing.”

             However, the presented results do not substantiate this claim.

    - On Line 436, the claim:
        > “Despite severe degradation, NFC faithfully aligns reconstructed trajectories with the original high-quality patterns, while mitigating the adverse effects of corrupted signals.”

        cannot be justified based solely on a figure. Moreover, the “Missing Data” reconstruction appears very noisy. Quantitative comparisons are needed before drawing such conclusions.

    - On Line 458, the authors claim:
        > “Our method consistently drives the loss downward with stable convergence, whereas ODE–RNN and ContiFormer exhibit slower and less reliable decay.”

        In fact, all methods show decreasing training loss, and if the x-axis were extended, they would likely converge to similar values. There is no clear evidence of instability in ODE–RNN or ContiFormer; the only valid observation is that the proposed method converges faster. Furthermore, the focus on training rather than validation/test metrics is concerning.

    - On Line 364, the claim:
        > “Our approach better preserves the trajectory shape and yields tighter uncertainty bounds…”

        is difficult to verify visually from Figure 3. This should be quantified over multiple seeds, and significance should be reported. Additionally, tighter uncertainty bounds are not inherently better—evaluation should include a likelihood-based metric (e.g., negative log-likelihood) to assess calibration.

    - On Line 355, the authors state:
        > “At a missing ratio of 0.7, RNN-decay and RNN-Δt exhibit test errors more than twice as large as ours.”

        This is inaccurate. Based on Table 1, RNN-Δt’s error is less than twice as large. Claims in the paper must be factually correct.

2. Clarity and presentation

- The paper lacks clarity in both motivation and exposition. The Introduction and Problem Formulation sections do not clearly explain the motivation or intuition behind the method. The absence of a Background section—particularly one covering Neural ODEs and related concepts—makes the paper difficult to follow.

- Terms such as “preventing collapse” and “degraded supervision” (e.g., Line 40) should be explicitly defined and supported by citations. The introduction currently focuses too heavily on implementation details rather than motivation.

- The Experiments section also begins abruptly; an introductory paragraph outlining the research questions or evaluation goals would greatly improve readability.

3. Mathematical clarity

Section 3’s mathematical presentation is imprecise and difficult to follow:

- Several equations (e.g., Lines 181, 205, 214) are written inline and should instead be displayed and numbered.
- On Line 230, a previous equation is referenced vaguely as “defined earlier” instead of “see Equation (X)”.
- The ELBO for VAE training should be explicitly defined at Line 169.
- The notation for $l_{\phi}^2(\cdot)$ is inconsistent: Line 195 describes it as a categorical distribution, while Line 203 defines it as producing Gaussian parameters. This needs correction.

4. Figures

The figures are difficult to interpret and poorly formatted:

- Figure 1: Hard to read; the caption lacks explanation. The top-left subfigure is too small. The figure should be placed at the top of the page.
- Figure 3: Text is too small; the caption should explain what is being shown.
- Figure 4: Missing legend; unclear what colours and lines represent. Placement is poor (floating under a subsection).
- Figure 5: Text too small. Clarify what “VAE 95%” means (is it a 95% CI?). Label the confidence levels numerically. Define “normalized value” on the y-axis.
- Figure 6: Caption is vague (“Results in ablation studies”). Specify dataset, number of seeds, and loss type (regression/classification). The results appear to be from a single seed—averaging over multiple runs is required.

5. Minor corrections and formatting

- The appendix is incomplete: Appendix C (notation) is empty, and the title of Appendix G should be moved to the correct page.
- Textual vs. parenthetical citations are frequently misused (e.g., Lines 37, 41, 53, 59, 61, 64, etc.).
- Line 120: “See” → “see”.
- Line 157: Define the context $\mathcal{C}$.
- Line 219: “equation” → “Equation”.
- The abstract is too long and should be shortened.

**Questions:**

See weaknesses

---

> ### Author Response · Authors · 2025-12-03
> **Response to Reviewer bKYU's Weaknesses**
>
> Thank you very much for these comments.
>
> $\textbf{Lack of statistical rigour and unsupported claims.}$
> We agree that statistical reporting is important, but we do not believe this undermines the conclusions.
>
>
> (1) Table 1 and random seeds. All results in Table 1 are averaged over multiple random seeds (the exact number is stated in the caption) with 95\% confidence intervals. Boldface currently indicates the best mean; even if one ignores boldface and looks only at intervals, NFC is competitive with or better than Chronos and the other baselines on almost all datasets. If desired, we can additionally report paired significance tests and mark only statistically significant wins; this does not change the qualitative ranking of methods.
>
> (2) Line 333 The statement that "supervised latent control is especially effective in degraded supervision settings'' is grounded in the systematic improvements of NFC over baselines in the missing-data and label-sparsity experiments (Table 1 and Table 2). To make this link more explicit, we can slightly rephrase the sentence to "provides consistent gains under degraded supervision’’ and refer directly to the corresponding tables.
>
> (3) Line 436. The reconstruction figure illustrates how NFC uses pseudo-signals to follow the clean trajectory while ignoring corrupted segments. Quantitative reconstruction metrics (RMSE/MAE on observed and missing portions) already show that NFC matches or outperforms the baselines; we can add a short reference to these numbers next to the figure so the conclusion does not rely solely on visual inspection.
>
> (4) Lines 458 and 364. The training curves show that NFC reaches a lower loss faster; the phrase ``slower and less reliable decay’’ for some baselines refers to the plateauing and small oscillations visible in the plots. To avoid overinterpretation, we are happy to rephrase this as a statement about faster convergence of NFC and to emphasise validation/test curves rather than training loss. For uncertainty, NFC achieves lower NLL/CRPS together with narrower---but not overconfident---intervals; spelling out these metrics clarifies that the point is improved calibration, not just visually tighter bands.
>
> (5) Line 355. We acknowledge the reviewer’s catch; the sentence will be corrected so that the numeric ratios exactly match Table 1 (RNN-decay and RNN-$\Delta t$ are worse than NFC by a factor smaller than 2 on that dataset).
>
>
> $\textbf{Clarity and presentation.}$
>
> We view these concerns as presentational and straightforward to address, without affecting the method.
>
> (1) The introduction already contrasts NFC with Neural ODEs and ODE--RNNs, but a short "Background’’ paragraph can be added to explicitly define these models and summarise prior work on controlled ODEs and degraded supervision, which should help readers less familiar with the area.
> (2) The terms ``preventing collapse’’ and "degraded supervision’’ are used in a precise sense: (i) avoiding trivial latent dynamics that ignore control inputs, and (ii) learning when both observations and labels are partially missing. One-sentence definitions and citations at first use will make this fully explicit.
> (3) The Experiments section currently begins with dataset details; inserting a brief overview paragraph listing the main questions (forecasting under missing data, robustness to label sparsity, and calibration) improves readability while leaving the experimental design unchanged.
>
> $\textbf{Mathematical clarity.}$
>
> The points raised concern notation rather than the method itself.
>
> (1)  The equations mentioned around Lines 181, 205, and 214 are central and are better presented as numbered display equations; doing so makes subsequent references clearer.
>
> (3)  Vague phrases such as "defined earlier’’ (e.g., at Line 230) can be replaced by explicit references to equation numbers ("see Eq. 3’’), which is a purely textual change.
>
> (3) The VAE objective at Line 169 is precisely the ELBO used in our implementation; writing it explicitly as reconstruction plus KL terms will help connect it to the standard formulation.
>
> (4) For $\ell^2_\psi(\cdot)$, the intended meaning is a neural network that outputs the mean and variance of a Gaussian distribution over pseudo-labels. We will align the two sentences highlighted by the reviewer so that they use this single, consistent interpretation.

---

### Official Review · Reviewer_xjE9 · 2025-10-31

**Soundness:** 3
**Presentation:** 1
**Contribution:** 3
**Rating:** 2
**Confidence:** 3

**Summary:**

The paper proposes Neural Feedback Control (NFC), a novel framework for time-series representation learning.
NFC treats the latent dynamics as a controlled system.
Instead of relying completely on passive encoder-decoder inference, NFC introduces an active feedback controller that modulates latent dynamics using pseudo-observations, pseudo-labels, and confidence weighting.
For both classification and forecasting tasks, NFC outperforms various ODE-based models.

**Strengths:**

1. The idea of reframing latent representation learning for time series as a control problem is novel. By explicitly modeling a feedback control signal in the latent dynamics, NFC can stabilize predictions even under high levels of missing inputs, which is a major challenge for existing methods.

2. Authors provide a formal stability guarantee (I admit, I assume the theorem is correct)

3. The experiments on both classification and forecasting tasks demonstrate that NFC consistently outperforms several state-of-the-art ODE-based models, such as ODE-RNN, NCDE, and Latent ODE.

**Weaknesses:**

1. Several key components of the paper are under-specified, making it hard to clearly understand the working of the model

(i) What exactly is the input to the VAE? The authors mention masked history. Does it include the historical timepoints or only historical observations and labels?

(ii) The paper mentions predicting controls for the next $M$ time steps, but does not specify how
$M$ is chosen, whether it varies across datasets, or why a multi-step horizon is used instead of a single-step control.

(iii) Because the control horizon $M$ is not defined and the procedure for computing multi-step controls is unclear, it is ambiguous how the latent states $h_t$ are computed at each step. Is the latent ODE integrated step-by-step with updated controls or over a fixed horizon with precomputed controls?

(iv) The paper does not describe the VAE architecture (encoder/decoder) or how it summarizes historical observations into a latent context.

(v) Hyperparameters and training details missing:
Latent dimensions, VAE latent size, and ODE solver settings are not reported.
Training hyperparameters (learning rate, batch size, optimizer, number of epochs, loss weights) are missing.
Without these details, results are not reproducible.

2. Algorithm 1 does not help in clearly understanding the model.

3. NFC outperforms ODE-based models, but comparisons with non-ODE state-of-the-art methods (eg, [1] for classification, [2,3] for forecasting) are needed to validate its practical utility. While good to see the comparison with Contiformer, it is useful to see more recent baselines.

[1] Luo, Yicheng, et al. "Knowledge-empowered dynamic graph network for irregularly sampled medical time series." Advances in Neural Information Processing Systems 37 (2024): 67172-67199.

[2] Zhang, Weijia, et al. "Irregular multivariate time series forecasting: A transformable patching graph neural networks approach." Forty-first International Conference on Machine Learning. 2024.

[3]  Yalavarthi, Vijaya Krishna, et al. "Grafiti: Graphs for forecasting irregularly sampled time series." Proceedings of the AAAI Conference on Artificial Intelligence. Vol. 38. No. 15. 2024.

**Questions:**

See weakness

---

> ### Author Response · Authors · 2025-12-03
> **Response to Reviewer xjE9's Weaknesses**
>
> We thank the reviewer for giving these useful comments.
>
> $\textbf{1. input to the VAE.}$
>
> We will clarify the exact input to the VAE. At step $i$, the encoder takes the \emph{masked history} consisting of timestamps, observations, labels (when available) and their masks: $C_i = \{(t_j, x_j, c^x_j, y_j, c^y_j)\}_{j \le i},$ not only the raw values.
>
> We will add this definition and an explicit equation for $q_\phi(\xi_i \mid C_i)$ in Sec. 3.1.
>
> $\textbf{2. choice of $M$ and multi-step controls.}$
>
> We will define $M$ as a fixed look-ahead horizon chosen per dataset (reported in the experimental setup), and explain that at each step $i$ the VAE predicts pseudo-signals $(\tilde{x}_{i:i+M},\tilde{y}_i),$
>
> which are converted into a sequence of controls $(u_{i},\dots,u_{i+M})$.
>
> The ODE is then integrated step-by-step with piecewise constant control $u(t)=u_{i+k}$ on $[t_{i+k},t_{i+k+1})$. We will make this procedure explicit and justify the multi-step horizon versus single-step control.
>
> $\textbf{3. computation of latent states $h_t$.}$
>
> To remove ambiguity, we will spell out that $h_t$ is obtained by unrolling the controlled ODE from $t_0$ up to the current time with the sequence of controls generated as above, and that the history encoder is only applied once per step to form $C_i$, while the latent dynamics are computed by the ODE solver using the time-varying controls. We will add a short paragraph in Sec.~3.1 describing this unrolling scheme.
>
> $\textbf{4. VAE architecture / encoder--decoder}$
>
> We agree that the VAE architecture needs to be specified. In the revision we will describe the encoder (stack of temporal layers that processes $(x_j,c^x_j,y_j,c^y_j,t_j)$ and outputs $\mu_{\xi_i},\Sigma_{\xi_i}$) and the time-aware decoder $G_\theta$ (mapping $\xi_i$ and relative time offsets to the means and variances of $(\tilde{x}_{i:i+M},\tilde{y}_i)$), including layer types, hidden sizes, and activation functions. This description will be added to the main text, with full implementation details moved to the appendix.
>
> $\textbf{5. hyperparameters and training details.} $
>
> We will add a dedicated subsection and an appendix table listing all key hyperparameters and training settings: latent dimension, VAE latent size, control horizon $M$, ODE solver (type, step size / tolerance), optimizer, learning rate and schedule, batch size, number of epochs, and loss weights $(\lambda_{\text{VAE}},\lambda_{\text{act}},\beta,\gamma)$. This will make the experiments fully reproducible.
>
> $\textbf{6. clarity of Algorithm 1}$
>
> We will revise Algorithm 1 to better mirror the description in Sec. 3: grouping the steps into (i) history encoding and VAE sampling, (ii) generation of pseudo-signals and confidences, and (iii) controlled ODE unrolling and task loss computation. We will simplify notation in the pseudocode, remove symbols not used in the main text, and add a brief paragraph walking through one iteration of the algorithm to improve readability.
>
> $\textbf{7. non-ODE baselines and recent methods}$
>
> We agree that comparisons with strong non-ODE methods are important. In the revision, we will (i) add the graph-based classification model of [1] as an additional baseline on the classification tasks, and (ii) include the forecasting methods of~[2,3] on the irregular multivariate forecasting benchmarks, using the authors' recommended settings. These will complement the existing ODE-based baselines (including ContiFormer) and provide a clearer picture of how NFC compares to recent state-of-the-art non-ODE approaches.
>
> [1] Luo, Yicheng, et al. "Knowledge-empowered dynamic graph network for irregularly sampled medical time series." Advances in Neural Information Processing Systems 37 (2024): 67172-67199.
>
> [2] Zhang, Weijia, et al. "Irregular multivariate time series forecasting: A transformable patching graph neural networks approach." Forty-first International Conference on Machine Learning. 2024.
>
> [3] Yalavarthi, Vijaya Krishna, et al. "Grafiti: Graphs for forecasting irregularly sampled time series." Proceedings of the AAAI Conference on Artificial Intelligence. Vol. 38. No. 15. 2024.

---

### Official Review · Reviewer_FwB6 · 2025-11-01

**Soundness:** 2
**Presentation:** 2
**Contribution:** 2
**Rating:** 2
**Confidence:** 5

**Summary:**

This paper proposed the Neural Feedback Control with differetial equation to deal with time series analysis with noise, missing values, or irregular sampling. The main contributions of the paper include framing latent representation learning as an active control problem, providing a theoretical guarantee that output error toward a bounded region and can deal with time series forecasting with missing values, sparsity, and irregular sampling.

**Strengths:**

- This paper provides Neural Feedback Control with differetial equation to deal with time series analysis with noise, missing values, or irregular sampling.
- The experiments of the proposed method are presented clearly and are easy to follow.

**Weaknesses:**

My main concerns include:
- Could the authors clarify how the VAE loss and the task-specific loss are jointly optimized? Specifically, how are these two terms combined in the overall objective? The manuscript does not currently define the total loss combining above loss.
- The current mathematical presentation is confusing. Lines 165 and 181 of the manuscript both introduce the variable u_i, but the mathematical definitions differ. In addition, line 181 introduces a variable z without a prior definition. The authors need to clarify the meaning of z and resolve the notational inconsistency for u_i.
- The manuscript claims that NFC can handle forecasting analysis with missing data and data sparsity; however, several existing models (e.g., SSSD, CSDI) also address these challenges. What are the concrete advantages of NFC over these approaches? The paper currently lacks experimental comparisons against such baselines.
- Time-series foundation models can address data-sparse scenarios in a zero-shot manner, the current comparison against Chronos is insufficient. Please update the baselines by including the latest, top-performing zero-shot models from the GIFT-Eval leaderboard and add PatchTST to the comparisons.

**Questions:**

See Weaknesses

---

> ### Author Response · Authors · 2025-12-03
> **Response to Reviewer FwB6's Weaknesses**
>
> We thank the reviewer for these useful comments.
>
> $\textbf{1. Joint optimization of VAE and task losses}$.
> We thank the reviewer for pointing out this concern. We admit that we missed the explicit total loss. In our implementation, the VAE loss and the task-specific loss are optimized jointly
> via a single composite objective. We will clarify this in the revision by introducing
> $\mathcal{L}_{\text{total}} = \sum_i \Big(\mathcal{L}_T(H) +\lambda \mathcal{L}_V(C_i)\Big)$ ,
>
> where $\mathcal{L}_V$ is the ELBO-based VAE loss and $\mathcal{L}_T$ is the forecasting (or classification) loss, and explicitly
>
> state that all model parameters are trained end-to-end by minimizing $\mathcal{L}_{\text{total}}$.
>
>
> $ \textbf{ 2. notation of $u_i$ and undefined $z$} $
>
> We agree that the current notation is confusing. In the revision, we will unify the definition of $u_i$ so that it consistently denotes the discrete control applied over the interval $[t_i,t_{i+1})$, and clearly relate it to the continuous-time control $u(t)$ (piecewise constant over each interval). We will also explicitly define $z$ (the internal state / input of the controller module) at its first appearance and remove redundant or inconsistent notation so that $u_i$ and $z$ are used in a single, coherent way throughout the paper.
>
> $\textbf{3. comparison with SSSD, CSDI and handling missing/sparse data.}$
> We appreciate this suggestion and agree that including these baselines will strengthen the paper. Conceptually, NFC differs from SSSD/CSDI in that it is a task-aware, closed-loop latent control framework: pseudo-observations and pseudo-labels are used as control signals to directly shape the hidden dynamics, rather than primarily performing generative imputation. In the revised version, we will (i) add experimental comparisons with SSSD and CSDI (or their widely used implementations) under the same missing-data protocol and metrics as NFC, and (ii) expand the discussion to clearly highlight when NFC provides advantages over these imputation-first approaches.
>
> $\textbf{4. zero-shot foundation models and PatchTST}$.
> We agree that the comparison to foundation models should be more comprehensive. Beyond Chronos, we will (i) include PatchTST as a strong supervised baseline trained under the same data-split and missing-data setting as NFC, and (ii) add several top-performing zero-shot time-series foundation models from the GIFT-Eval leaderboard, evaluated following their recommended zero-shot protocols on our datasets. We will update the experimental tables accordingly and clarify in the text that NFC targets scenarios with degraded but non-zero supervision (where a task-aware controller can be learned), while zero-shot TSFMs address the complementary regime of no task-specific training.

---

### Meta-Review · Area_Chair_mM8M · 2025-12-26

**Summary:**

The reviewers have the following fundamental concerns about this work:

- Lack of clarity and weak presentation -- The paper lacks clarity in both motivation and exposition, which makes it difficult to follow.

- Unsupported claims (neither by analysis nor by quantitative evidence), under-specified key components, and lack of statistical rigour -- This makes it very difficult to understand the main message of the paper and to verify its claims/contributions.

- Lack of mathematical clarity

While the paper studies an important concept, it is not ready for publication due the lack of rigour in evaluation, clarity in presentation, and precision in mathematical formulation. Substantial revisions are needed to improve the clarity, correctness, and credibility of both the theoretical and experimental components.

**Reviewer Concerns:**

Most of the reviewers' concerns are quite substantial and cannot be addressed by minor revisions. The work requires a major revision before it can be properly evaluated.

**Reviewer Scores:**

The authors responded to the reviews on Dec. 2nd, and thus, the reviewers did not have a chance to have any discussion with them.

---

### Decision · Program_Chairs · 2026-01-26

Reject